# Paediatric COVID-19 mortality: a database analysis of the impact of health resource disparity

Eva Miranda Marwali [1,2] Aria Kekalih,[3] Saptadi Yuliarto,[4] Dyah Kanya Wati,[5] Muhammad Rayhan,[1] Ivy Cerelia Valerie,[5] Hwa Jin Cho,[6] Waasila Jassat,[7] Lucille Blumberg,[7] Maureen Masha,[7] Calum Semple,[2,8] Olivia V Swann,[9] Malte Kohns Vasconcelos [10] Jolanta Popielska,[11] Srinivas Murthy,[12] Robert A Fowler,[13] Anne-Marie Guerguerian,[14] Anca Streinu-Cercel,[15] Mohan Dass Pathmanathan,[16] Amanda Rojek,[2,17] Christiana Kartsonaki,[2] Bronner P Gonçalves,[2] Barbara Wanjiru Citarella [2] Laura Merson,[2] Piero L Olliaro,[2] Heidi Jean Dalton,[18] on behalf of International Severe Acute Respiratory and emerging Infection Consortium (ISARIC) Clinical Characterization Group Investigators

EMM and HJD are joint senior authors.

For numbered affiliations see end of article.

**Correspondence to**
Dr Eva Miranda Marwali; eva. marwali@pjnhk.go.id

## ABSTRACT

**Background** The impact of the COVID-19 pandemic on paediatric populations varied between high-income countries (HICs) versus low-income to middle-income countries (LMICs). We sought to investigate differences in paediatric clinical outcomes and identify factors contributing to disparity between countries.

**Methods** The International Severe Acute Respiratory and Emerging Infections Consortium (ISARIC) COVID-19 database was queried to include children under 19 years of age admitted to hospital from January 2020 to April 2021 with suspected or confirmed COVID-19 diagnosis. Univariate and multivariable analysis of contributing factors for mortality were assessed by country group (HICs vs LMICs) as defined by the World Bank criteria.

**Results** A total of 12 860 children (3819 from 21 HICs and 9041 from 15 LMICs) participated in this study. Of these, 8961 were laboratory-confirmed and 3899 suspected COVID-19 cases. About 52% of LMICs children were black, and more than 40% were infants and adolescent. Overall in-hospital mortality rate (95% CI) was 3.3% [=(3.0% to 3.6%), higher in LMICs than HICs (4.0% (3.6% to 4.4%) and 1.7% (1.3% to 2.1%), respectively). There were significant differences between country income groups in intervention profile, with higher use of antibiotics, antivirals, corticosteroids, prone positioning, high flow nasal cannula, non-invasive and invasive mechanical ventilation in HICs. Out of the 439 mechanically ventilated children, mortality occurred in 106 (24.1%) subjects, which was higher in LMICs than HICs (89 (43.6%) vs 17 (7.2%) respectively). Pre-existing infectious comorbidities (tuberculosis and HIV) and some complications (bacterial pneumonia, acute respiratory distress syndrome and myocarditis) were significantly higher in LMICs compared with HICs. On multivariable analysis, LMIC as country income group was associated with increased risk of mortality (adjusted HR 4.73 (3.16 to 7.10)).

## WHAT IS ALREADY KNOWN ON THIS TOPIC

⇒ Recent systematic reviews identified a potentially higher paediatric mortality per population from COVID-19 in low-income to middle-income countries (LMICs) compared with high-income countries (HICs) but concluded that heterogeneity of published studies limits firm conclusions. Correspondingly, lethality of other acute respiratory infections has been shown to be higher in LMIC.

## WHAT THIS STUDY ADDS

⇒ By using harmonised data collection tools of a study population of over 12 000 children, this study can directly compare inpatient management and outcomes in HIC and LMIC. Analysis finds higher mortality in LMIC, although a lower proportion receive intensive care unit admission and ventilation prior to death. Disparity in access to care and lack of available advanced medical therapies are highlighted and provide areas for collaborative efforts between clinicians, administrators and likely government groups to improve outcomes in LMIC.

## HOW THIS STUDY MIGHT AFFECT RESEARCH, PRACTICE OR POLICY

⇒ While intensified by the pandemic, a lack of adequate resources to care for children with acute respiratory infections in LMIC is likely a general concern that requires allocation of resources. Reducing the gap in our ability to care for sick children in LMICs versus HICs will inevitably improve global outcomes during both pandemic and interpandemic periods.

**Conclusion** Mortality and morbidities were higher in LMICs than HICs, and it may be attributable to differences in patient demographics, complications and access to supportive and treatment modalities.

## INTRODUCTION

The clinical presentation, severity and outcomes of acute COVID-19 are different in children compared with adults. While a higher proportion of children are asymptomatic or less severely ill than in many adult reports, severe manifestations do occur.[1] While cardiac compromise in the form of multisystem inflammatory syndrome in children (MIS-C) is often described, acute respiratory distress syndrome (ARDS) and other organ dysfunction also occurs in children.[2 3] The risk factors for severe disease in in paediatrics are incompletely understood.[4] Furthermore, there is a lack of global data to improve understanding of the COVID-19 burden in children who live in low-income to middle-income countries (LMICs) versus those in high-income countries (HICs) sites.[5]

One systematic review that summarised the difference in paediatric COVID-19 morbidity and mortality in HICs and LMICs has been published.[6] However, most studies' samples analysed fewer than 100 patients, and over half the data came from the USA and China. Furthermore, due to heterogeneous reporting of data in the included studies, the authors were limited in their ability to pool or compare data on clinical features and outcomes. A recent scoping review reported that ethnicity and socioeconomic condition were under-represented in COVID-19 epidemiological studies.[7] We aimed to investigate the differences in survival of paediatric patients in LMICs and HICs and the factors that may contribute to such differences between regions.

## METHODS

Data were collected from hospitalised patients under 19 years of age with confirmed or clinically suspected COVID-19 between 1 January 2020 and 31 March 2021 admitted to institutions across the globe contributing to the International Severe Acute Respiratory and Emerging Infections Consortium (ISARIC) database according to ISARIC/WHO Clinical Characterisation Protocol for Severe Emerging Infection.[1 8] Fields for analysis were extracted from the complete dataset gathered from Research Electronic Data Capture (V.8.11.11, Vanderbilt University, Nashville, Tennessee, USA).[8]

Variables of interest were classified into four domains: comorbidities, presenting signs and symptoms, complications and treatments. Comorbidity was defined as any history of pre-existing medical conditions that were not otherwise related to COVID-19 natural history and reported at admission date. Complications referred to any medical condition detected during patients' stay that was not present at admission. Therapies included were drugs, oxygen and use of other treatments such as invasive mechanical ventilation (IMV). Ethnicity was collapsed into five categories (black or African American, white, Asian, mixed/others and missing/unknown) following the Centers for Disease Control and Prevention National Health Interview Survey glossary.[9] Country

**Table 1** List of participating countries and number of admitted subjects (n=12 860)

| Income group | Country name | n | % |
|---|---|---|---|
| High-income country | UK | 3099 | 24.1 |
| High-income country | Poland | 304 | 2.4 |
| High-income country | Canada | 164 | 1.3 |
| High-income country | Spain | 87 | 0.7 |
| High-income country | Germany | 25 | 0.2 |
| High-income country | USA | 22 | 0.2 |
| High-income country | Ireland | 21 | 0.2 |
| High-income country | France | 18 | 0.1 |
| High-income country | Australia | 17 | 0.1 |
| High-income country | Chile | 16 | 0.1 |
| High-income country | Israel | 12 | 0.1 |
| High-income country | Netherlands | 11 | 0.1 |
| High-income country | Italy | 8 | 0.1 |
| High-income country | Greece | 4 | 0.0 |
| High-income country | Belgium | 3 | 0.0 |
| High-income country | Portugal | 3 | 0.0 |
| High-income country | Bolivia | 1 | 0.0 |
| High-income country | Kuwait | 1 | 0.0 |
| High-income country | Norway | 1 | 0.0 |
| High-income country | New Zealand | 1 | 0.0 |
| High-income country | Saudi Arabia | 1 | 0.0 |
| Lower-middle income country | South Africa | 7621 | 59.3 |
| Lower-middle income country | Malaysia | 627 | 4.9 |
| Lower-middle income country | Malawi | 296 | 2.3 |
| Lower-middle income country | Romania | 135 | 1.0 |
| Lower-middle income country | Colombia | 112 | 0.9 |
| Lower-middle income country | Indonesia | 73 | 0.6 |
| Lower-middle income country | Peru | 49 | 0.4 |
| Lower-middle income country | Pakistan | 35 | 0.3 |
| Lower-middle income country | India | 24 | 0.2 |
| Lower-middle income country | Honduras | 21 | 0.2 |
| Lower-middle income country | Argentina | 18 | 0.1 |
| Lower-middle income country | Mexico | 11 | 0.1 |
| Lower-middle income country | Brazil | 10 | 0.1 |
| Lower-middle income country | Nepal | 6 | 0.0 |
| Lower-middle income country | Russia | 3 | 0.0 |

income groups were dichotomised into HICs and LMICs according to the latest World Bank classification.[10]

Date of admission was defined as the date of hospitalisation. Descriptive statistics were described as frequencies (n) and proportions for categorical data, mean±SD or median (IQR) for continuous data, and number of available data (N) for each variable. Demographic characteristics, comorbidities, complications and treatments were compared between country income groups using $\chi^2$ or Fisher's exact test as indicated. Kaplan-Meier survival curves were plotted and compared using the log-rank test. Multivariable Cox proportional hazards regression models were fitted to identify mortality predictors. In-hospital survival analysis was performed to obtain 28-day and 90-day survival rates. Intensive care unit (ICU) admission and IMV requirement served as proxies for morbidity. Time-to-event analyses were performed to identify morbidity and mortality defining periods: (1) from hospital admission until ICU admission, (2) ICU admission to first intubation, (3) ICU admission to ICU discharge or death, (4) ICU discharge to hospital discharge or death. A p value of <0.05 was considered statistically significant. Statistical analyses were performed using SPSS V.25 (IBM Corp).

## RESULTS

There were 12 860 children, originating from 36 countries, 15 of which (41.7%) were LMICs and 21 (58.3%) HICs (table 1). Seventy per cent (n=9041) of participants were from LMICs and 29.7% (n=3819) cases were from HICs. The majority of participants were contributed from 670 cites in South Africa (59.3%) and 346 sites in the UK (24.1%), followed by Malaysia (4.9%), Poland (2,4%) and Malawi (2.3%) (figure 1 and table 1). COVID-19 status was laboratory confirmed in 69.7%; this rate was higher in HICs (73.0%) in comparison with that in LMICs (68.3%) (table 2).

Adolescents aged between 12 and 17 years were the largest age group in LMICs (25.6%) and overall (24.0%), followed by infants younger than 1 year of age (21%). Infants were the largest group in HICs (29.4%). Males (50.8%) and females (49.2%) were similarly represented. Black or African-American participants formed more than one-third of the study population and 52% of LMICs children (table 2).

In total, 425 (3.3%) participants died. The mortality rate was higher in in LMICs compared with HICs (4% vs 1.7%) (table 2). Reported mortality in the UK was 52 children (1.2%), which does not differ significantly (p=0.98) from the rest of HICs. Mortality in South Africa, numbering 268 children (3.5%), differed significantly (p<0.001) from 93 (6.5%) mortality in the rest of LMICs. The total admissions and mortality rates across the study period based on country groups (UK, South Africa and other countries) were presented in figure 2. The curves showed bimodal with peaks of mortality rates coinciding with admissions during July 2020 and January 2021. Available case data in early 2020 was limited compared with later time points. In contrast to South Africa and other countries, mortality rate in the UK did not rise significantly with the first and second wave of case admissions during April–May 2020 and October 2020–January 2021.

Children in LMICs had significantly greater and earlier mortality (adjusted HR (aHR) (95% CI) 4.73 (3.16 to 7.10), p<0.001). The 28-day and 90-days survival among all participants were 96.7% (10 339/10 692) and 96.5% (10 642/11 024), respectively. Survival was higher at both 28 (98.3% (3049/3101)) and 90 days (98.1% (3137/3199)) in HICs than LMICs (96.0% (7290/7591) and 95% (7505/7825), respectively).

The availability of information on comorbidities varied between countries of origin. The prevalence of several comorbidities was significantly higher in HICs including chronic neurological disease, seizures, diabetes and chronic cardiac disease. Infectious diseases such as tuberculosis and HIV/AIDS were significantly higher in LMICs (table 3).

The most common presenting symptoms overall include fever (20%), cough (16.1%) and shortness of breath (10.5%). Between-group's comparisons showed that cough (HIC 14.0% vs LMIC 26.1%), shortness of breath (9.2% vs 16,4%), runny nose (4.2% vs 8.6%), loss of smell or taste (0.7% vs 2.9%), anorexia (0.1% vs 0.8%) and inability to walk (0.1% vs 0.2%) were more common among patients in LMICs.

Complications during hospitalisation are shown in table 4. Patients in LMICs had more bacterial and cryptogenic pneumonia and ARDS, as well as other organ involvement such as brain with stroke or heart with cardiac arrest. The rates of complications were generally higher than that of HICs, except for cardiac arrhythmia and disseminated intravascular coagulation in which HICs rates were statistically higher.

The two most commonly administered therapies were antibiotics in 41.2% of participants, followed by corticosteroids in 11.4%. Antivirus was administered in 5.3% of subjects, with higher percentage observed in HIC, especially of remdesivir, which is more than twice that of LMIC. Adjunctive and supportive treatments were generally performed more often in HICs. No participants in LMICs were treated with extracorporeal membrane

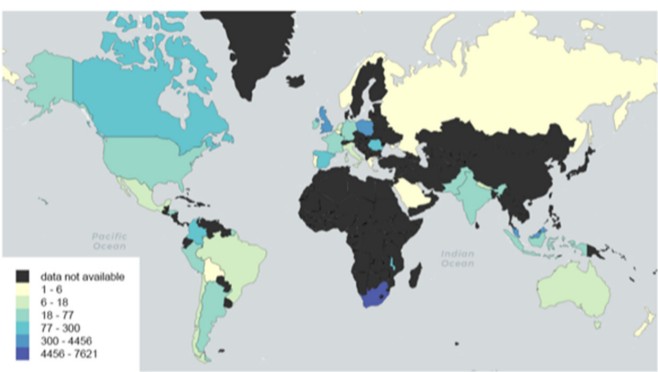

**Figure 1** Distribution of participants by country of origin.

**Table 2** Baseline characteristics of all study participants (n=12 860)

| | | Country income | | | | | | |
|---|---|---|---|---|---|---|---|---|
| | | HICs (n=3819) | | LMICs (n=9041) | | Total | | |
| | | n | % | n | % | n | % | P value |
| Age* | <1 year | 1124 | 29.4 | 1574 | 17.4 | 2698 | 21.0 | <0.001 |
| | 1–<5 years | 707 | 18.5 | 1887 | 20.9 | 2594 | 20.2 | |
| | 5–<12 years | 796 | 20.8 | 1621 | 17.9 | 2417 | 18.8 | |
| | 12–<17 years | 770 | 20.2 | 2311 | 25.6 | 3081 | 24.0 | |
| | ≥17 years | 422 | 11.1 | 1648 | 18.2 | 2070 | 16.1 | |
| Sex* | Male | 2026 | 53.1 | 4497 | 49.8 | 6523 | 50.8 | 0.002 |
| | Female | 1788 | 46.9 | 4537 | 50.2 | 6325 | 49.2 | |
| Ethnicity* | Black or African-American | 223 | 5.8 | 4747 | 52.5 | 4970 | 38.6 | <0.001 |
| | Missing and unknown | 1221 | 32.0 | 3509 | 38.8 | 4730 | 36.8 | |
| | White | 1624 | 42.5 | 186 | 2.1 | 1810 | 14.1 | |
| | Asian | 561 | 14.7 | 168 | 1.9 | 729 | 5.7 | |
| | Mixed and others | 190 | 5.0 | 431 | 4.8 | 621 | 4.8 | |
| COVID-19 status* | Confirmed | 2786 | 73 | 6175 | 68.3 | 8961 | 69.7 | <0.001 |
| | Suspect | 1033 | 27 | 2866 | 31.7 | 3899 | 30.3 | |
| Outcome* | Discharged | 3423 | 89.6 | 7967 | 88.1 | 11 390 | 88.6 | <0.001 |
| | Still in hospital | 140 | 3.7 | 357 | 3.9 | 497 | 3.9 | |
| | Transferred | 192 | 5.0 | 356 | 3.9 | 548 | 4.3 | |
| | Death | 64 | 1.7 | 361 | 4.0 | 425 | 3.3 | |

*Significant with p value <0.05 using $\chi^2$ test.
HICs, high-income countries; LMICs, low- to middle-income countries.

oxygenation, as compared with 10 participants in HICs (table 5).

Participants in LMICs were most often admitted to the ICU within the first day of admission, with those who died being admitted earlier than survivors (LMICs vs HICs: 0 (0–1) vs 0 (0–2.5) day respectively, p=0.03). While time to IMV was not significantly different for survivors versus

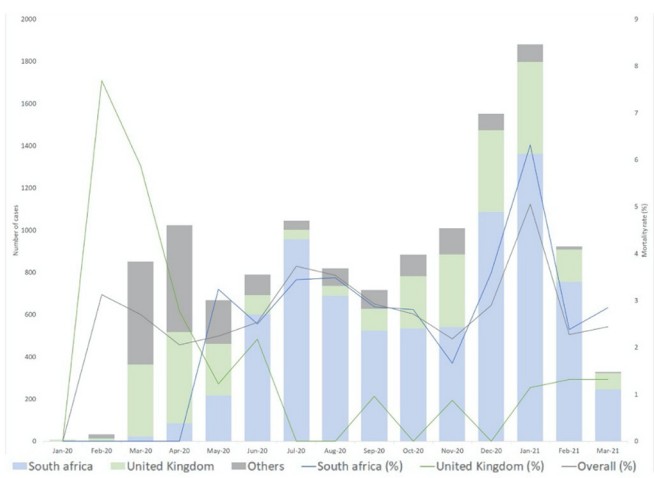

**Figure 2** Total COVID-19 participants (bar) and mortality (line) rate trend through out data collection period.

non-survivors in either LMIC or HICs, most received IMV in the first day of admission and all within a maximum of 3.5 days. Time from ICU admission to ICU discharge or death was significantly shorter in survivors versus non-survivors, although non survivors in HICs had longer stays than in LMICs (3 (1–6) vs 9.5 (4.5–19.2) days respectively in HIC and 0 (0–4) vs 4 (0.7–7.2) days respectively in LMIC, p<0.05). Mortality was significantly higher (p<0.001) in LMICs compared with HICs for participants who received IMV (43.6% vs 7.3%) or who required ICU admission (16.7% vs 3.5). Nearly 2.3% and 6.1% participants died without ICU and/or IMV support in LMICs in comparison with the respective 1.3% and 1.4% in HICs (table 6).

Multivariable analysis was done to evaluate factors associated with mortality. Significant risk factors found were: aged <1 year (aHR (95% CI)=1.80 (1.01 to 3.22)), low-middle income group (4.73 (3.16 to 7.10)), comorbidities such as chronic kidney disease (3.74 (2.20 to 6.35)) or cardiac disease (2.42 (1.50 to 3.91)) and invasive mechanical ventilator requirement (3.46 (2.27 to 5.28)) or exposure to antibiotics (2.07 (1.34 to 3.22)). The use of antiviral agents (aHR=0.55 (0.32 to 0.96)) was the only factor inversely associated with mortality (figure 3).

**Table 3** Comparison of comorbidities between country income groups

| Country income | | | | | | | | | | |
|---|---|---|---|---|---|---|---|---|---|---|
| | HICs | | | LMICs | | | Total | | | |
| | N | n | % | N | n | % | N | n | % | P value |
| Chronic neurological disorder* | 2588 | 191 | 7.4 | 902 | 14 | 1.6 | 3490 | 205 | 5.9 | 0.001 |
| Seizure* | 2490 | 111 | 4.5 | 899 | 16 | 1.8 | 3389 | 127 | 3.7 | 0.001 |
| Smoking | 2091 | 82 | 3.9 | 1509 | 46 | 3.0 | 3600 | 128 | 3.6 | 0.215 |
| Diabetes* | 2592 | 95 | 3.7 | 4468 | 149 | 3.3 | 7060 | 244 | 3.5 | 0.001 |
| Chronic cardiac disease* | 2587 | 141 | 5.5 | 3578 | 48 | 1.3 | 6165 | 189 | 3.1 | 0.001 |
| Obesity* | 2438 | 90 | 3.7 | 1736 | 39 | 2.2 | 4174 | 129 | 3.1 | 0.001 |
| Tuberculosis* | 290 | – | – | 3649 | 102 | 2.8 | 3939 | 102 | 2.6 | 0.001 |
| Chronic haematological disease* | 2475 | 77 | 3.1 | 899 | 9 | 1.0 | 3374 | 86 | 2.5 | 0.001 |
| HIV/AIDS* | 2562 | 1 | <0.01 | 4382 | 142 | 3.2 | 6944 | 143 | 2.1 | 0.001 |
| Rare diseases and inborn errors of metabolism | 1823 | 36 | 2.0 | 0 | – | – | 1823 | 36 | 2.0% | 0.715 |
| Malnutrition | 2539 | 35 | 1.4 | 901 | 26 | 2.9% | 3440 | 61 | 1.8% | 0.412 |
| Malignant neoplasm* | 2571 | 64 | 2.5 | 4164 | 27 | 0.6% | 6735 | 91 | 1.4% | 0.001 |
| Rheumatological disorder* | 2491 | 45 | 1.8 | 898 | 2 | 0.2% | 3389 | 47 | 1.4% | 0.001 |
| Chronic kidney disease* | 2596 | 59 | 2.3 | 4147 | 28 | 0.7 | 6743 | 87 | 1.3 | 0.001 |
| Liver disease | 2623 | 12 | 0.5 | 902 | – | – | 3525 | 12 | 0.3 | 0.866 |

*Significant with p-value <0.05 using Chi-square test.
HICs, high-income countries; LMICs, low-income to middle-income countries; N, denominators; n, total patients with comorbidities.

## DISCUSSION

We present a large international cohort of children hospitalised with COVID-19. We found that mortality was significantly higher in LMICs in comparison with HICs. Disparate care patterns were also observed, with patients in LMICs reported to receive most adjunctive and supportive therapies less frequently than patients in HICs. While these findings may represent differences in practice, they may also represent variation in available supports for children based on income status of the country. Such disparities have been described in adult COVID-19 patients, but limited data exist for children. Prior reports have focused on specific aspects of illness such as infection or cardiac dysfunction, have included small cohorts of children or are limited to certain countries or regions.[11–20]

While the findings may be criticised as mainly representing data from two countries, the UK and South Africa (SA), these countries are good examples of HICs and LMICs. Statistical analysis showed no significant difference of mortality between UK and the rest of HICs. Although mortality in SA was significantly lower than the rest of LMICs, both mortalities from SA and non-SA LMICs were significantly higher than HICs group. Low number of subjects from non-SA LMICs was disproportionate to that of SA, thus conclusion can not be drawn from observed difference in mortality between them. Furthermore, as data supplied from the UK and South Africa comes from national COVID-19 research databases recruiting from a high number of sites in the UK and South Africa, and in this sense may be more representative of country income differences, as opposed to

enrolment of single sites (eg, a national referral centre) in different countries. The inclusion of children from many other countries, although relatively small cohorts form each country in comparison, does allow understanding of care patterns in areas around the world.

More participants from HICs could be admitted to ICU and received IMV than LMICs patients. While the small numbers available for analysis in some categories limit our confidence in these findings, in LMICs they do given IMV and dying within shorter periods of time than HICs. Not only were children in LMICs hospitalised with COVID-19 more likely to die, they were also shown to die earlier in their hospitalisation. Despite possible confounding effects from missing data relating to severity of illness at presentation (vital signs, organ failure scores), a positive association between LMICs and mortality were consistently observed in analysis of children admitted to the ICU and those receiving IMV.

Several independent risk factors for mortality were identified in addition to country economic group. Mortality was lowest for patients aged between 1 and 5 years and higher among patients of age <1 or >5 years. This finding confirmed the U-shaped mortality pattern shown in several other reports, although infancy is not always recognised as a risk factor in small studies.[1 21 22] Comorbidities such as chronic kidney and cardiac diseases were also shown to be independent risk factors as reported by others.[19 23 24] These risk factors were more prevalent in HICs and thus did not seem to be associated with the high mortality rates noted in LMICs, but this may reflect underdiagnosis possibly due to lack of diagnostic resources in some LMICs.[20] Higher rates of certain comorbidities, many

**Table 4** Comparison of complications between country income groups

| Country income | HICs | | | LMICs | | | Total | | | |
|---|---|---|---|---|---|---|---|---|---|---|
| | N | n | % | N | n | % | N | n | % | P value |
| Bacterial pneumonia* | 3497 | 115 | 3.3 | 1107 | 102 | 9.2 | 4604 | 217 | 4.7 | <0.001 |
| ARDS* | 3153 | 31 | 1.0 | 4972 | 267 | 5.4 | 8125 | 298 | 3.7 | <0.001 |
| AKI | 3624 | 110 | 3.0 | 1410 | 49 | 3.5 | 5034 | 159 | 3.2 | 0.477 |
| Seizure | 3627 | 103 | 2.8 | 1412 | 39 | 2.8 | 5039 | 142 | 2.8 | 0.958 |
| Pleural effusion | 3511 | 82 | 2.3 | 1110 | 36 | 3.2 | 4621 | 118 | 2.6 | 0.118 |
| Myocarditis and pericarditis* | 2858 | 42 | 1.5 | 427 | 22 | 5.2 | 3285 | 64 | 1.9 | <0.001 |
| Bronchiolitis | 3620 | 57 | 1.6 | 1412 | 21 | 1.5 | 5032 | 78 | 1.6 | 0.922 |
| Cardiac arrhythmia* | 3635 | 70 | 1.9 | 1407 | 13 | 0.9 | 5042 | 83 | 1.6 | 0.017 |
| Endocarditis* | 692 | 5 | 0.7 | 487 | 14 | 2.9 | 1179 | 19 | 1.6 | 0.008 |
| Cardiac arrest* | 3535 | 19 | 0.5 | 1412 | 55 | 3.9 | 4947 | 74 | 1.5 | <0.001 |
| DIC* | 3498 | 48 | 1.3 | 4668 | 58 | 1.2 | 8166 | 104 | 1.3 | 0.850 |
| Meningitis and encephalitis* | 3621 | 16 | 0.4 | 1413 | 20 | 1.4 | 5034 | 36 | 0.7 | <0.001 |
| Pneumothorax | 3530 | 15 | 0.4 | 1109 | 9 | 0.8 | 4639 | 24 | 0.5 | 0.185 |
| Stroke/cerebrovascular complication† | 3527 | 7 | 0.2 | 1109 | 11 | 1.0 | 4636 | 18 | 0.4 | 0.001 |
| Pulmonary embolism | 1928 | 4 | 0.2 | 293 | 2 | 0.7 | 2221 | 6 | 0.3 | 0.182 |
| Cardiac ischaemia | 3510 | 7 | 0.2 | 1110 | 4 | 0.4 | 4620 | 11 | 0.2 | 0.309 |
| Myocardial infarction | 312 | 1 | 0.3 | 298 | – | – | 610 | 1 | 0.2 | 1.000 |
| Sepsis | – | – | – | 3862 | 8 | 0.2 | 3862 | 8 | 0.2 | – |
| DVT | 1606 | 1 | 0.1 | 3 | – | – | 1609 | 1 | 0.1 | 1.000 |
| COP† | 3398 | – | – | 1110 | 5 | 0.5 | 4508 | 5 | 0.1 | 0.001 |

*Significant with p-value <0.05 using $\chi^2$ test.
†Fisher's exact test.
AKI, acute kidney injury; ARDS, acute respiratory distress syndrome; COP, cryptogenic organising pneumonia; DIC, disseminated intravascular coagulation; DVT, deep vein thrombosis; HICs, high-income countries; LMICs, low-income to middle-income countries; n, total patients with complications; N, denominators.

infectious in nature, is another possible cause of the relationship between LMICs and mortality. Chronic respiratory failure has been associated with death in COVID-19 adult patients, with some evidence that this is occurs in paediatric cases as well. Our data also provided information on tuberculosis, which has not specifically identified as comorbidity in children with COVID-19 in other reports. Similarly, data on the impact of HIV in children is sparse, and our review finds this to be an important risk factor and more prevalent in LMICs.[23 25]

More patients receiving antiviral therapy were found in HICs versus LMICS. In fact, remdesivir, which is recommended for severe hospitalised COVID-19, was used in exceptionally lower percentage of LMIC subjects. We can only speculate that period of this study occurred when evidence based on antiviral efficacy was still scarce especially in children, or it may indicate lack of access to drug or lack familiarity with recommendations.[26] The recommendations for antiviral use in children with severe COVID-19 from the National Institutes of Health have suggested use for patients over the age of 12 years; this recommendation would not have applied to infants who had a higher risk of mortality in our study.[27 28] Further investigation of the impact of antivirals in children of all ages should be considered. The efficacy and the cost benefit of these expensive medications in resource-limited sites are needed; if valuable, improving access should then be at the core of discussions.

The use of other therapies also highlight the differences between LMICs and HICs. There was less use of many adjunct therapies associated with outcomes in adult studies such as prone positioning, high flow nasal cannula, antibiotics and steroids in LMIC sites. Whether these differences were the result of lack of availability of therapies or other regional factors cannot be determined, but it seems likely that limitation to access may influence practice. The high rate of mortality in patients outside the ICU and who died without IMV suggests that limitations to ICU beds or ventilators in LMICs likely play a

**Table 5** Treatment profile of study participants

| | Country income | | | | | | | | | |
|---|---|---|---|---|---|---|---|---|---|---|
| | HICs | | | LMICs | | | Total | | | |
| | N | n | % | N | n | % | N | n | % | P value |
| Antibiotics* | 3685 | 2189 | 59.4 | 4349 | 1122 | 25.8 | 8034 | 3311 | 41.2 | <0.001 |
| Corticosteroid* | 3663 | 564 | 15.4 | 4905 | 417 | 8.5 | 8568 | 981 | 11.4 | <0.001 |
| HFNC* | 3380 | 282 | 8.3 | 3967 | 147 | 3.7 | 7347 | 429 | 5.8 | <0.001 |
| IMV* | 3620 | 235 | 6.5 | 4684 | 204 | 4.3 | 8304 | 439 | 5.3 | <0.001 |
| Antivirus* | 3672 | 271 | 7.4 | 4341 | 172 | 4.0 | 8013 | 443 | 5.3 | <0.001 |
| Remdesivir | 3672 | 96 | 2.6 | 4341 | 17 | 0.4 | 8013 | 113 | 1.4 | |
| Neuraminidase inhibitor | 3672 | 32 | 0.9 | 4341 | 23 | 0.5 | 8013 | 113 | 55 | |
| Inotropic/vasopressor* | 3451 | 201 | 5.8 | 4961 | 174 | 3.5 | 8412 | 375 | 4.4 | <0.001 |
| Prone positioning* | 3532 | 64 | 1.8 | 4647 | 39 | 0.8 | 8179 | 103 | 1.2 | <0.001 |
| Anticoagulant | 3819 | 50 | 1.3 | 9041 | 88 | 1.0 | 12 860 | 138 | 1.1 | 0.091 |
| NIV* | 3627 | 53 | 1.5 | 4966 | 17 | 0.3 | 8593 | 70 | 0.8 | <0.001 |
| RRT | 3573 | 19 | 0.5 | 4334 | 27 | 0.6 | 7907 | 46 | 0.6 | 0.084 |
| ECMO† | 3608 | 10 | 0.3 | 1086 | – | – | 4694 | 10 | 0.2 | <0.001 |

*Significant with p value <0.05 using $\chi^2$ test.
†Fisher's exact test.
ECMO, extracorporeal membrane oxygenation; HFNC, high-flow nasal cannula; HICs, high-income countries; IMV, invasive mechanical ventilation; LMICs, low-income to middle-income countries; N, denominators; n, total patients with treatments; NIV, non-invasive ventilation; RRT, renal replacement therapy.

large role in the excess death rates reported as compared with HICs.[29 30]

The higher prevalence of complications of respiratory disease such as ARDS, bacterial and cryptogenic organising pneumonia, and the impact of organ dysfunction outside the lung such as increased rates of myocarditis, pericarditis, endocarditis, meningitis, encephalitis, stroke and cardiac arrest observed in LMICs are also likely factors in the high death rates. Patients with MIS-C were not specifically reported in the time period of this report.

Our study has the strength of a common reporting format in participating centres around the world. We describe a relatively large number of children and are able to provide both comparisons of patient characteristics and outcomes and evaluate risk factors for outcomes using common definitions. Limitation of this study includes a predominance of patients from one LMICs and one HICs, South Africa and UK, potentially limiting the generalisability of our findings to all countries. In addition, we did not adjust for pandemic era. Inevitably, we have missing data for a number of variables, including comorbidities, which limits the effective sample size of analyses examining relationships with patient characteristics and outcomes. Lack of data on nutritional status of children on each group, which may explain disparity between country income groups, was another limitation of the study. Moreover, considerable proportion of non-confirmed cases also limits the impact of this study on public health policy.

In conclusion, we found many differences in characteristics, treatments and outcomes among children from LMICs and HICs with infants had higher death rates than other children. Patients less frequently receive IMV and other supportive therapies in LMICs, which likely represents disparities in access to healthcare that influence outcomes. Reducing the gap in our ability to care

**Table 6** Mortality based on ICU admission and IMV use

| Mortality based on country income | | | | | | | | | | |
|---|---|---|---|---|---|---|---|---|---|---|
| | | HICs | | | LMICs | | | Total | | |
| | | N | n | % | N | n | % | N | n | % |
| ICU* | Yes | 679 | 24 | 3.5 | 1037 | 173 | 16.7 | 1716 | 197 | 11.5 |
| | No | 3140 | 40 | 1.3 | 8004 | 188 | 2.3 | 11 144 | 228 | 2.0 |
| IMV* | Yes | 235 | 17 | 7.2 | 204 | 89 | 43.6 | 439 | 106 | 24.1 |
| | No | 3385 | 47 | 1.4 | 4480 | 272 | 6.1 | 7865 | 319 | 4.0 |

*Significant with p value <0.05 using $\chi^2$ test .
HICs, high-income countries; ICU, intensive care unit; IMV, invasive mechanical ventilation; LMICs, low-income to middle-income countries; N, denominator (total patients with or without ICU/ IMV); n, total mortality.

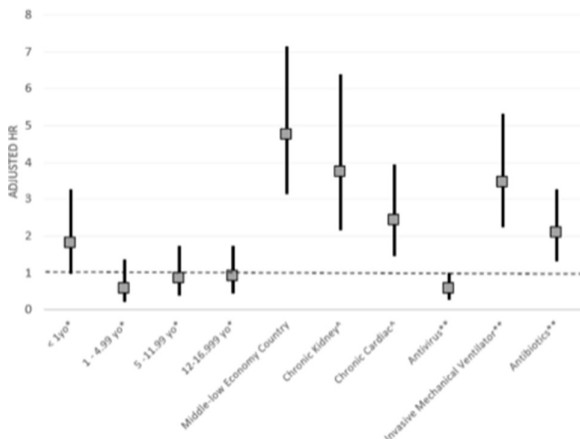

**Figure 3** Adjusted HR and 95% CI of mortality risk factors in all participants.

for sick children in LMICs versus HICs will inevitably improve global outcomes during both pandemic and interpandemic periods.

**Author affiliations**

[1]Department of Pediatric Cardiac Intensive Care, National Cardiovascular Center Harapan Kita, Jakarta, Indonesia

[2]International Severe Acute Respiratory and emerging Infection Consortium (ISARIC), Pandemic Sciences Institute, University of Oxford, Oxford, UK

[3]Department of Community Medicine, Faculty of Medicine, Universitas Indonesia, Jakarta, Indonesia

[4]Department of Pediatrics, Faculty of Medicine, Universitas Brawijaya, Saiful Anwar Hospital, Malang, Jawa Timur, Indonesia

[5]Department of Pediatrics, Faculty of Medicine, Universitas Udayana, Sanglah Hospital, Denpasar, Bali, Indonesia

[6]Division of Pediatric Cardiology and Pediatric Intensive Care, Department of Pediatrics, Chonnam National University Hospital, Gwangju, Korea (the Republic of)

[7]National Institute for Communicable Diseases, Johannesburg, South Africa

[8]NIHR Health Protection Research Unit in Emerging and Zoonotic Infections, Institute of Infection Veterinary and Ecological Sciences, University of Liverpool, Liverpool, UK

[9]Centre for Medical Informatics, The University of Edinburgh Usher Institute of Population Health Sciences and Informatics, Edinburgh, UK

[10]Centre for Neonatal and Paediatric Infection, St George's University of London, London, UK

[11]Warsaw's Hospital for Infectious Diseases, Medical University of Warsaw, Warszawa, Mazowieckie, Poland

[12]Department of Pediatrics, Division of Critical Care, The University of British Columbia, Vancouver, British Columbia, Canada

[13]Interdepartmental Division of Critical Care Medicine, University of Toronto, Toronto, Ontario, Canada

[14]Department of Critical Care Medicine, Neurosciences and Mental Health Program, Faculty of Medicine, University of Toronto, The Hospital for Sick Children, Toronto, Ontario, Canada

[15]National Institute of Infectious Diseases Prof Dr Matei Bals, Bucuresti, Romania

[16]National Institutes of Health, Ministry of Health Malaysia, Putrajaya, Wilayah Persekutuan, Malaysia

[17]The Royal Melbourne Hospital, Parkville, Victoria, Australia

[18]Inova Fairfax Medical Center, Inova, Falls Church, Arizona, USA

**Acknowledgements** This work was made possible by the UK Foreign, Commonwealth and Development Office and Wellcome [215091/Z/18/Z, 222410/Z/21/Z, 225288/Z/22/Z and 220757/Z/20/Z]; the Bill & Melinda Gates Foundation [OPP1209135]; the philanthropic support of the donors to the University of Oxford's COVID-19 Research Response Fund (0009109); CIHR Coronavirus Rapid Research Funding Opportunity OV2170359 and the coordination in Canada by Sunnybrook Research Institute; endorsement of the Irish Critical Care- Clinical Trials Group, co-ordination in Ireland by the Irish Critical Care- Clinical Trials Network at University College Dublin and funding by the Health Research Board of Ireland [CTN-2014-12]; the COVID clinical management team, AIIMS, Rishikesh, India; the COVID-19 Clinical Management team, Manipal Hospital Whitefield, Bengaluru, India; Cambridge NIHR Biomedical Research Centre; the dedication and hard work of the Groote Schuur Hospital Covid ICU Team and supported by the Groote Schuur nursing and University of Cape Town registrar bodies coordinated by the Division of Critical Care at the University of Cape Town; the Liverpool School of Tropical Medicine and the University of Oxford; the dedication and hard work of the Norwegian SARS-CoV-2 study team and the Research Council of Norway grant no 312780, and a philanthropic donation from Vivaldi Invest A/S owned by Jon Stephenson von Tetzchner; Imperial NIHR Biomedical Research Centre; the Comprehensive Local Research Networks (CLRNs) of which PJMO is an NIHR Senior Investigator (NIHR201385); Innovative Medicines Initiative Joint Undertaking under Grant Agreement No. 115523 COMBACTE, resources of which are composed of financial contribution from the European Union's Seventh Framework Programme (FP7/2007- 2013) and EFPIA companies, in-kind contribution; Stiftungsfonds zur Förderung der Bekämpfung der Tuberkulose und anderer Lungenkrankheiten of the City of Vienna, Project Number: APCOV22BGM; Italian Ministry of Health "Fondi Ricerca corrente–L1P6" to IRCCS Ospedale Sacro Cuore–Don Calabria; Australian Department of Health grant (3273191); Gender Equity Strategic Fund at University of Queensland, Artificial Intelligence for Pandemics (A14PAN) at University of Queensland, the Australian Research Council Centre of Excellence for Engineered Quantum Systems (EQUS, CE170100009), the Prince Charles Hospital Foundation, Australia; grants from Instituto de Salud Carlos III, Ministerio de Ciencia, Spain; Brazil, National Council for Scientific and Technological Development Scholarship number 303953/2018- 7; the Firland Foundation, Shoreline, Washington, USA; the French COVID cohort (NCT04262921) is sponsored by INSERM and is funded by the REACTing (REsearch & ACtion emergING infectious diseases) consortium and by a grant of the French Ministry of Health (PHRC n°20-0424); a grant from foundation Bevordering Onderzoek Franciscus; the South Eastern Norway Health Authority and the Research Council of Norway; Institute for Clinical Research (ICR), National Institutes of Health (NIH) supported by the Ministry of Health Malaysia; preparedness work conducted by the Short Period Incidence Study of Severe Acute Respiratory Infection; the U.S. DoD Armed Forces Health Surveillance Division, Global Emerging Infectious Diseases Branch to the U.S Naval Medical Research Unit No. TWO (NAMRU-2) (Work Unit #: P0153_21_N2). These authors would like to thank Vysnova Partners, Inc. for the management of this research project. The Lao-Oxford-Mahosot Hospital-Wellcome Trust Research Unit is funded by the Wellcome Trust. This work uses data provided by patients and collected by the NHS as part of their care and support #DataSavesLives. The data used for this research were obtained from ISARIC4C. We are extremely grateful to the 2648 frontline NHS clinical and research staff and volunteer medical students who collected these data in challenging circumstances; and the generosity of the patients and their families for their individual contributions in these difficult times. The COVID-19 Clinical Information Network (CO-CIN) data was collated by ISARIC4C Investigators. Data and Material provision was supported by grants from: the National Institute for Health Research (NIHR; award CO-CIN-01), the Medical Research Council (MRC; grant MC_PC_19059), and by the NIHR Health Protection Research Unit (HPRU) in Emerging and Zoonotic Infections at University of Liverpool in partnership with Public Health England (PHE), (award 200907), NIHR HPRU in Respiratory Infections at Imperial College London with PHE (award 200927), Liverpool Experimental Cancer Medicine Centre (grant C18616/A25153), NIHR Biomedical Research Centre at Imperial College London (award ISBRC-1215-20013), and NIHR Clinical Research Network providing infrastructure support. We also acknowledge the support of Jeremy J Farrar and Nahoko Shindo.

**Collaborators** International Severe Acute Respiratory and Emerging Infections Consortium Clinical Characterisation Group: Sheryl Ann Abdukahil, Nurul Najmee Abdulkadir, Laurent Abel, Lara Absil, Subhash Acharya, Diana Adriao, Younes Ait Tamlihat, Ernita Akmal, Eman Al Qasim, Tala Al-dabbous, Beatrice Alex, Kévin Alexandre, Huda Alfoudri, Kazali Enagnon Alidjnou, Clotilde Allavena, Nathalie Allou, Rita Alves, Joana Alves Cabrita, Maria Amaral, Nur Amira, Phoebe Ampaw, Claire Andrejak, Andrea Angheben, Francois Angoulvant, Severine Ansart, Sivanesen Anthonidass, Carlos Alexandre Antunes de Brito, Ardiyan Apriyana, Yaseen Arabi, Carolline Araujo, Patrick Archambault, Jean-Benoit Arlet, Christel Arnold-Day, Elise Artaud-Macari, Diptesh Aryal, Muhammad Ashraf, Jean Baptiste Assie, Amirul Asyraf, Anika Atique, AM Udara Lakshan Attanyake, Johann Auchabie, Hugues Aumaitre, Adrien Auvet, Laurène Azemar, Cecile Azoulay, Benjamin Bach, Delphine Bachelet, Claudine Badr, J Kenneth Baillie, Nazreen Abu Bakar, Mohanaprasanth Balakrishnan, Valeria Balan, Firouze Bani-Sadr, Renata Barbalho, Wendy S Barclay, Saef Umar Barnett, Audrey Barrelet, Marie Bartoli, Joaquín Baruch, Romain Basmaci, Muhammad Fadhli Hassin Basri, Jules Bauer, Diego Fernando Bautista Rincon, Abigail Beane, Alexandra Bedossa, Ker Hong Bee, Husna Begum, Sylvie

Behilill, Albertus Beishuizen, Anna Beltrame, Marine Beluze, Nicolas Benech, Lionel Eric Benjiman, Dehbia Benkerrou, Delphine Bergeaud, Jose Luis Bernal Sobrino, Giulia Bertoli, Simon Bessis, Sybille Bevilcaqua, Karine Bezulier, Nowneet Kumar Bhat, Krishna Bhavsar, Farah Nadiah Bidin, Felwa Bin Humaid, Mohd Nazlin Bin Kamarudin, Francois Bissuel, Laurent Bitker, Jonathan Bitton, Mathieu Blot, Lucille Blumberg, Laetitia Bodenes, Debby Bogaert, Anne-Helene Boivin, Isabela Bolaños, Pierre-Adrien Bolze, Francois Bompart, Diogo Borges, Raphael Borie, Hans Martin Bosse, Elisabeth Botelho-Nevers, Lila Bouadma, Olivier Bouchaud, Sabelline Bouchez, Dounia Bouhmani, Damien Bouhour, Kevin Bouiller, Laurence Bouillet, Camile Bouisse, Anne-Sophie Boureau, Maude Bouscambert, Aurore Bousquet, Jason Bouziotis, Bianca Boxma, Marielle Boyer-Besseyre, Fernando Augusto Bozza, Axelle Braconnier, Cynthia Braga, Timo Brandenburger, Kathy Brickell, Marjolein Brusse-Keizer, Polina Bugaeva, Marielle Buisson, Erlina Burhan, Aidan Burrell, Ingrid G Bustos, Denis Butnaru, Andre Cabie, Eder Caceres, Cyril Cadoz, Rui Caetano Garces, Jose Andres Calvache, Valentine Campana, Paul Campbell, Pauline Caraux-Paz, Chiara Simona Cardellino, Sofia Cardoso, Filipe Cardoso, Filipa Cardoso, Nicolas Carlier, Thierry Carmoi, Marie-Christine Carret, Francois Martin Carrier, Gail Carson, Maire-Laure Casanova, Mariana Cascao, Jose Casimiro, Nidyanara Castanheira, Guylaine Castor-Alexandre, Francois-Xavier Catherine, Paolo Cattaneo, Roberta Cavalin, Minerva Cervantes-Gonzalez, Anissa Chair, Catherine Chakveatze, Adrienne Chan, Meera Chand, Christelle Chantalat Auger, Jean-Marc Chapplain, Julie Chas, Anjellica Chen, Matthew Pellan Cheng, Antoine Cheret, Thibault Chiarabini, Julian Chica, Suresh Kumar Chidambaram, Leong Chin Tho, Catherine Chirouze, Bernard Cholley, Marie-Charlotte Chopin, Ting Soo Chow, Hiu Jian Chua, Jonathan Chua, Barbara Wanjiru Citarella, Emma Clarke, Sara Clohisey, Alexandra Coelho, Megan Coles, Gwenhael Colin, Pamela Combs, Marie Connor, Anne Conrad, Graham S Cooke, Mary Copland, Hugues Cordel, Amanda Corley, Sabine Cornelis, Alexander Daniel Cornet, Arianne Joy Corpuz, Grégory Corvaisier, Emma Costigan, Camille Couffignal, Sandrine Couffin-Cadiergues, Roxane Courtois, Stephanie Cousse, Rachel Cregan, Sabine Croonen, Claudina Cruz, Juan Luis Cruz Bermúdez, Jaime Cruz Rojo, Elodie Curlier, Paula Custodio, Ana da Silva Filipe, Charlene Da Silveira, Andrew Dagens, Heidi Dalton, Jo Dalton, Juliana Damas, Nick Daneman, Corinne Daniel, Emmanuelle A Dankwa, Jorge Dantas, Etienne De Montmollin, Rafael Freitas de Oliveira Franca, Thushan de Silva, Peter de Vries, Jillian Deacon, Alexa Debard, Marie-Pierre Debray, Nathalie DeCastro, William Dechert, Lauren Deconninck, Romain Decours, Eve Defous, Isabelle Delacroix, Eric Delaveuve, Karen Delavigne, Christelle Delmas, Pierre Delobel, Corine Delsing, Elisa Demonchy, Emmanuelle Denis, Dominique Deplanque, Pieter Depuydt, Diane Descamps, Mathilde Desvallees, Santi Dewayanti, Pathik Dhanger, Alpha Diallo, Sylvain Diamantis, André Dias, Rodrigo Diaz, Kevin Didier, Jean-Luc Diehl, Wim Dieperink, Jerome Dimet, Vincent Dinot, Fara Diop, Alphonsine Diouf, Yael Dishon, Felix Djossou, Annemarie B Docherty, Arjen M Dondorp, Maria Donnelly, Christl A Donnelly, Chloe Donohue, Celine Dorival, Eric D'Ortenzio, James Joshua Douglas, Nathalie Dournon, Tom Drake, Murray Dryden, Murray Dryden, Claudio Duarte Fonseca, Vincent Dubee, Francois Dubos, Alexandre Ducancelle, Paul Dunand, Jake Dunning, Bertrand Dussol, Xavier Duval, Sim Choon Ean, Mohammed El Sanharawi, Brigitte Elharrar, Philippine Eloy, Isabelle Enderle, Chan Chee Eng, Ilka Engelmann, Vincent Enouf, Olivier Epaulard, Martina Escher, Hélène Esperou, Catarina Espirito Santo, Marina Esposito-Farese, Joao Estevao, Manuel Etienne, Nadia Ettalhaoui, Mirjam Evers, Marc Fabre, Isabelle Fabre, Amna Faheem, Arabella Fahy, Cameron J Fairfield, Pedro Faria, Ahmed Farooq, Hanan Fateena, Karine Faure, Raphael Favory, Jorge Fernandes, Marília Andreia Fernandes, Francois-Xavier Ferrand, Eglantine Ferrand Devouge, Joana Ferrao, Mario Ferraz, Bernardo Ferreira, Nicolas Ferriere, Celine Ficko, Claudia Figueiredo-Mello, Thomas Flament, Clara Flateau, Tom Fletcher, Victor Fomin, Tatiana Fonseca, Patricia Fontela, Simon Forsyth, Erwan Fourn, Robert A Fowler, John F Fraser, Christophe Fraser, Marcela Vieira Freire, Ana Freitas Ribeiro, Stephanie Fry, Valerie Gaborieau, Rostane Gaci, Jean-Charles Gagnard, Amandine Gagneux-Brunon, Carrol Gamble, Yasmin Gani, Noelia Garcia Barrio, Esteban Garcia-Gallo, Denis Garot, Valerie Garrait, Nathalie Gault, Alexandre Gaymard, Eva Geraud, Louis Gerbaud Morlaes, Nuno Germano, Jade Ghosn, Carlo Giaquinto, Jess Gibson, Tristan Gigante, Morgane Gilg, Guillermo Giordano, Michelle Girvan, Valérie Gissot, Daniel Glikman, Petr Glybochko, Geraldine Goco, François Goehringer, Siri Goepel, Jean-Christophe Goffard, Jin Yi Goh, Marie Gominet, Bronner P Gonçalves, Patricia Gordon, Isabelle Gorenne, Laure Goubert, Cécile Goujard, Tiphaine Goulenok, Pascal Granier, Christopher A Green, Courtney Greene, William Greenhalf, Segolène Greffe, Fiona Griffiths, Albert Groenendijk, Anja Grosse Lordemann, Heidi Gruner, Yusing Gu, Jeremie Guedj, Martin Guego, Dewi Guellec, Anne-Marie Guerguerian, Daniela Guerreiro, Romain Guery, Anne Guillaumot, Laurent Guilleminault, Thomas Guimard, Marieke Haalboom, Daniel Haber, Hannah Habraken, Ali Hachemi, Nadir Hadri, Fakhir Haidri, Matthew Hall, Sophie Halpin, Ansley Hamer, Rebecca Hamidfar, Lim Yuen Han, Rashan Haniffa, Kok Wei Hao,

Hayley Hardwick, Ewen M Harrison, Janet Harrison, Lars Heggelund, Ross Hendry, Maxime Hentzien, Diana Hernandez, Astarini Hidayah, Rupert Higgins, Samuel Hinton, Hikombo Hitoto, Antonia Ho, Yi Bin Ho, Alexandre Hoctin, Isabelle Hoffmann, Wei Han Hoh, Peter Horby, Ikram Houas, Catherine L Hough, Jimmy Ming-Yang Hsu, Jean-Sebastien Hulot, Samreen Ijaz, Arfan Ikram, Hajnal-Gabriela Illes, Patrick Imbert, Hugo Inacio, Yun Sii Ing, Elias Iosifidis, Sarah Isgett, Tiago Isidoro, Nadiah Ismail, Margaux Isnard, Danielle Jaafar, Salma Jaafoura, Julien Jabot, Clare Jackson, Nina Jamieson, Pierre Jaquet, Waasila Jassat, Coline Jaud-Fischer, Stephane Jaureguiberry, Florence Jego, Anilawati Mat Jelani, Ong Yiaw Joe, Mark Joseph, Cedric Joseph, Swosti Joshi, Merce Jourdain, Philippe Jouvet, Dafsah Juzar, Ouifiya Kafif, Florentia Kaguelidou, Neerusha Kaisbain, Thavamany Kaleesvran, Sabina Kali, Muhammad Aisar Ayadi Kamaluddin, Zul Amali Che Kamaruddin, Nadiah Kamarudin, Kong Yeow Kang, Pratap Karpayah, Todd Karsies, Christiana Kartsonaki, Anant Kataria, Kevin Katz, Seán Keating, Yvelynne Kelly, Sadie Kelly, Kalynn Kennon, Younes Kerroumi, Sharma Keshav, Antoine Khalil, Coralie Khan, Irfan Khan, Quratul Ain Khan, Sushil Khanal, Abid Khatak, Saye Khoo, Ryan Khoo, Denisa Khoo, Khor How Kiat, Peter Kiiza, Antoine Kimmoun, Detlef Kindgen-Milles, Paul Klenerman, Rob Klont, Gry Kloumann Bekken, Stephen R Knight, Robin Kobbe, Chamira Kodipily, Malte Kohns Vasconcelos, Arsène Kpangon, Vinothini Krishnan, Ganesh Kumar, Bharath Kumar Tirupakuzhi Vijayaraghavan, Pavan Kumar Vecham, Vinod Kumar, Neurinda Permata Kusumastuti, Demetrios Kutsogiannis, Marie Lachatre, Marie Lacoste, Marie Lagrange, Fabrice Laine, Olivier Lairez, Antonio Lalueza, Marc Lambert, Marie Langelot-Richard, Vincent Langlois, Marina Lanza, Cédric Laouénan, Samira Laribi, Delphine Lariviere, Stephane Lasry, Odile Launay, Didier Laureillard, Yoan Lavie-Badie, Andy Law, Teresa Lawrence, Minh Le, Clement Le Bihan, Cyril Le Bris, Georges Le Falher, Lucie Le Fevre, Quentin Le Hingrat, Marion Le Marechal, Soizic Le Mestre, Gwenael Le Moal, Vincent Le Moing, Herve Le Nagard, Paul Le Turnier, Ema Leal, Marta Leal Santos, Todd C Lee, James Lee, Jennifer Lee, Heng Gee Lee, Biing Horng Lee, Yi Lin Lee, Gary Leeming, Laurent Lefebvre, Bénédicte Lefebvre, Benjamin Lefevre, Sylvie LeGac, Jean-Daniel Lelievre, Francois Lellouche, Adrien Lemaignen, Veronique Lemee, Anthony Lemeur, Ha Sha Lene, Jenny Lennon, Marc Leone, Quentin Lepiller, Francois-Xavier Lescure, Olivier Lesens, Mathieu Lesouhaitier, Yves Levy, Bruno Levy, Claire Levy-Marchal, Erwan L'Her, Gianluigi Li Bassi, Geoffrey Liegeon, Wei Shen Lim, Kah Chuan Lim, Bruno Lina, Lim Lina, Guillaume Lingas, Sylvie Lion-Daolio, Samantha Lissauer, Marine Livrozet, Patricia Lizotte, Navy Lolong, Leong Chee Loon, Diogo Lopes, Anthony L Loschner, Paul Loubet, Bouchra Loufti, Guillame Louis, Lee Lee Low, Marije Lowik, Jia Shyi Loy, Jean Christophe Lucet, Carlos Lumbreras Bermejo, Liem Luong, Dominique Luton, Ruth Lyons, Oryane Mabiala, Sara Machado, Moise Machado, Gabriel Macheda, Hashmi Madiha, Guillermo Maestro de la Calle, Rafael Mahieu, Sophie Mahy, Mylene Maillet, Thomas Maitre, Nadia Malik, Fernando Maltez, Denis Malvy, Victoria Manda, Jose M Mandei, Laurent Mandelbrot, Julie Mankikian, Edmund Manning, Aldric Manuel, Ceila Maria Sant Ana Malaque, Daniel Marino, Samuel Markowicz, Laura Marsh, John Marshall, Celina Turchi Martelli, Guillaume Martin-Blondel, Martin Martinot, Ana Martins, Caroline Martins Rego, Eva Miranda Marwali, Marsilla Marzukie, David Maslove, Sabina Mason, Sobia Masood, Basri Mat Nor, Moshe Matan, Daniel Mathieu, Mathieu Mattei, Romans Matulevics, Laurence Maulin, Thierry Mazzoni, Colin McArthur, Anne McCarthy, Sarah E McDonald, Kenneth A McLean, Paul McNally, Cecile Mear-Passard, Ogechukwu Menkiti, Kusum Menon, France Mentre, Alexander J Mentzer, Noemie Mercier, Emmanuelle Mercier, Antoine Merckx, Mayka Mergeay-Fabre, Laura Merson, António Mesquita, Agnes Meybeck, Alison M Meynert, Vanina Meysonnier, Amina Meziane, Mehdi Mezidi, Céline Michelanglei, Isabelle Michelet, Nor Arisah Misnan, Tahira Jamal Mohamed, Nik Nur Eliza Mohamed, Asma Moin, Agostinho Monteiro, Claudia Montes, Shona C Moore, Sarah Moore, Lina Morales Cely, Lucia Moro, Ben Morton, Hugo Mouquet, Clara Mouton Perrot, Julien Moyet, Caroline Mudara, Ng Yong Muh, Dzawani Muhamad, Jimmy Mullaert, Karl Erik Muller, Daniel Munblit, Marlène Murris, Srinivas Murthy, Himed Musaab, Himasha Muvindi, Gugapriyaa Muyandy, Nadege Neant, Nikita Nekliudov, Raul Neto, Anthony Nghi, Duc Nguyen, Alistair Nichol, Nurul Amani Mohd Noordin, Marion Noret, Nurul Faten Izzati Norharizam, Lisa Norman, Mahdad Noursadeghi, Karolina Nowicka, Saad Nseir, Nurnaningsih Nurnaningsih, Elsa Nyamankolly, Annmarie O'Callaghan, Katie O'Hearn, Agnieszka Oldakowska, Piero L Olliaro, David S Y Ong, Jee Yan Ong, Wilna Oosthuyzen, Peter Openshaw, Saijad Orakzai, Claudia Milena Orozco-Chamorro, Linda O'Shea, Siti Zubaidah Othman, Nadia Ouamara, Rachida Ouissa, Eric Oziol, Maïder Pagadoy, Justine Pages, Massimo Palmarini, Prasan Kumar Panda, Lai Hui Pang, Nathalie Pansu, Aurelie Papadopoulos, Rachael Parke, Melissa Parker, Jeremie Pasquier, Bruno Pastene, Mohan Dass Pathmanathan, Juliette Patrier, Mical Paul, Christelle Paul, Jorge Paulos, William A Paxton, Jean-François Payen, Kalaiarasu Peariasamy, Miguel Pedrera Jiménez, Florent Peelman, Nathan Peiffer-Smadja, Vincent Peigne, Ithan D Peltan, Rui Pereira, Daniel Perez, Thomas Perpoint, Vincent Pestre, Michele Petrovic,

Ventzislava Petrov-Sanchez, Gilles Peytavin, Scott Pharand, Walter Picard, Olivier Picone, Carola Pierobon, Djura Piersma, Carlos Pimentel, Raquel Pinto, Catarina Pires, Isabelle Pironneau, Lionel Piroth, Riinu Pius, Laurent Plantier, Hon Shen Png, Julien Poissy, Ryadh Pokeerbux, Maria Pokorska-Spiewak, Georgios Pollakis, Diane Ponscarme, Jolanta Popielska, Douwe F Postma, Diana Povoas, Sebastien Preau, Jean-Charles Preiser, Mark G Pritchard, Gamage Dona Dilanthi Priyadarshani, Oriane Puechal, Vilmaris Quinones-Cardona, Victor Quiros Gonzalez, Mohammed Quraishi, Christian Rabaud, Marie Rafiq, Rozanah Abd Rahman, Ahmad Kashfi Haji Ab Rahman, Giri Shan Rajahram, Nagarajan Ramakrishnan, Jose Ramalho, Ahmad Afiq Ramli, Blandine Rammaert, Grazielle Viana Ramos, Christophe Rapp, Aasiyah Rashan, Thalha Rashan, Menaldi Rasmin, Cornelius Rau, Tharmini Ravi, Stanislas Rebaudet, Sarah Redl, Brenda Reeve, Attaur Rehman, Jonathan Remppis, Martine Remy, Hongru Ren, Hanna Renk, Anne-Sophie Resseguier, Matthieu Revest, Oleksa Rewa, Luis Felipe Reyes, David Richardson, Laurent Richier, Siti Nurul Atikah Ahmad Ridzuan, Ana L Rios, Asgar Rishu, Patrick Rispal, Karine Risso, Stephanie Roberts, David L Robertson, Olivier Robineau, Paola Rodari, Pierre-Marie Roger, Emmanuel Roilides, Amanda Rojek, Juliette Romaru, Mélanie Roriz, Manuel Rosa-Calatrava, Michael Rose, Andrea Rossanese, Bénédicte Rossignol, Patrick Rossignol, Stella Rousset, Carine Roy, Benoit Roze, Desy Rusmawatiningtyas, Clark D Russell, Musharaf Sadat, Valla Sahraei, Maximilien Saint-Gilles, Stephane Sallaberry, Charlotte Salmon Gandonniere, Helene Salvator, Olivier Sanchez, Vanessa Sancho-Shimizu, Zulfiqar Sandhu, Pierre-Francois Sandrine, Oana Sandulescu, Shirley Sarfo-Mensah, Benjamine Sarton, Egle Saviciute, Parthena Savvidou, Yen Tsen Saw, Arnaud Scherpereel, Marion Schneider, Janet T Scott, James Scott-Brown, Nicholas Sedillot, Jaganathan Selvanayagam, Mageswari Selvarajoo, Caroline Semaille, Malcolm G Semple, Rasidah Bt Senian, Eric Senneville, Tania Sequeira, Ary Serpa Neto, Pablo Serrano Balazote, Ellen Shadowitz, Syamin Asyraf Shahidan, Mohammad Shamsah, Shaikh Sharjeel, Catherine A Shaw, Victoria Shaw, Ashraf Sheharyar, Mohiuddin Shiekh, Sally Shrapnel, Nassima Si Mohammed, Ng Yong Siang, Jeanne Sibiude, Louise Sigfrid, Benedict Sim Lim Heng, Karisha Sivam, Sue Smith, Morgane Snacken, Tze Vee Soh, Joshua Solomon, Tom Solomon, Agnes Sommet, Rima Song, Tae Song, Azlan Mat Soom, Albert Sotto, Edouard Soum, B P Sanka Ruwan Sri Darshana, Shiranee Sriskandan, Sarah Stabler, Ymkje Stiensstra, Adrian Streinu-Cercel, Anca Streinu-Cercel, David Stuart, Jacky Y Suen, Charlotte Summers, Deepashankari Suppiah, Andrey Svistunov, Sarah Syahrin, Jaques Sztajnbok, Shirin Tabrizi, Fabio S Taccone, Lysa Tagherset, Shahdattul Mawarni Taib, Ewa Talarek, Kim Keat Tan, Yan Chyi Tan, Coralie Tardivon, Pierre Tattevin, M Azhari Taufik, Richard S Tedder, Tze Yuan Tee, Joao Teixeira, Marie-Capucine Tellier, Sze Kye Teoh, François Teoule, Olivier Terrier, Nicolas Terzi, Hubert Tessier-Grenier, Alif Adlan Mohd Thabit, Zhang Duan Tham, Suvintheran Thangavelu, Vincent Thibault, Simon-Djamel Thiberville, Benoit Thill, Jananee Thirumanickam, David Thomson, Emma C Thomson, Surain Raaj Thanga Thurai, Ryan S Thwaites, Paul Tierney, Peter S Timashev, Jean-Francois Timsit, Noémie Tissot, Jordan Zhien Yang Toh, Sia Loong Tonnii, Marta Torre, Margarida Torres, Tony Trapani, Theo Treoux, Cecile Tromeur, Tiffany Trouillon, Jeanne Truong, Christelle Tual, Sarah Tubiana, Jean-Marie Turmel, Lance C W Turtle, PG Ishara Udayanga, Andrew Udy, Timothy M Uyeki, Luis Val-Flores, Amelie Valran, Marcel van den Berge, Job van der Palen, Paul van der Valk, Peter Van der Voort, Sylvie Van Der Werf, Jarne Van Hattem, Carolien van Netten, Ilonka van Veen, Noemie Vanel, Shoban Raj Vasudayan, Charline Vauchy, Shaminee Veeran, Aurelie Veislinger, Annelies Verbon, James Vickers, Jose Ernesto Vidal, Cesar Vieira, Benoit Visseaux, Harald Vonkeman, Fanny Vuotto, Noor Hidayu Wahab, Suhaila Abdul Wahab, Nadirah Abdul Wahid, Steve Webb, Jia Wei, Katharina Weil, Tan Pei Wen, Sanne Wesselius, Murray Wham, Nicole White, Paul Henri Wicky, Aurelie Wiedemann, Evert-Jan Wils, Calvin Wong, Xin Ci Wong, Yew Sing Wong, Teck Fung Wong, Gan Ee Xian, Lim Saio Xian, Kuan Pei Xuan, Siti Rohani Binti Mohd Yakop, Yazdan Yazdanpanah, Nicholas Yee Liang Hing, Cecile Yelnik, Chian Hui Yeoh, Stephanie Yerkovich, Hodane Yonis, Obada Yousif, Saptadi Yuliarto, Marion Zabbe, Masliza Zahid, Maram Zahran, Nor Zaila Binti Zaidan, Maria Zambon, Konrad Zawadka, Nurul Zaynah, Hiba Zayyad, David Zucman.

**Contributors** The author contributions are as follows: study conception and design: EMM, AK, SY, DKW, MR, ICV, HJC, WJ, LB, MM, CS, OS, MKV, JP, SM, RAF, A-MG, AS-C, MDP, AR, CK, BPG, BWC, LM, PLO, and HJD; coordination and collection of data: EMM, AK, SY, DKW, MR, ICV, HJC, WJ, LB, MM, CS, OS, MKV, JP, SM, RAF, A-MG, AS-C, MDP, AR, CK, BPG, BWC, LM, PLO, and HJD ; statistical analysis or interpretation of data: EMM, AK, SY, DKW, MR, ICV, HJC, WJ, LB, MM, CS, OS, MKV, JP, SM, RAF, A-MG, AS-C, MDP, AR, CK, BPG, BWC, LM, PLO and HJD; drafting the manuscript: EMM, MR, ICV and HJD; critical review and revision of the manuscript for important intellectual insight: EMM, AK, SY, DKW, MR, ICV, HJC, WJ, LB, MM, CS, OS, MKV, JP, SM, RAF, A-MG, AS-C, MDP, AR, CK, BPG, BWC, LM, PLO and HJD; study supervision: EMM, BWC and LM; guarantor: EMM, BWC, LM and HJD. The authors drafted the manuscript from important intellectual viewpoints

and approved the final version. Furthermore, all authors agreed to be accountable for all aspects of the work in ensuring that questions related to the accuracy or integrity of any part of the work are appropriately investigated and resolved. The corresponding author confirmed that all authors meet authorship criteria according to ICMJE.

**Funding** National Institute for Health Research (NIHR), UK Medical Research Council, Wellcome Trust, Department for International Development, Bill & Melinda Gates Foundation, EU Platform for European Preparedness Against (Re-)emerging Epidemics, NIHR Health Protection Research Unit (HPRU) in Emerging and Zoonotic Infections at University of Liverpool, NIHR HPRU in Respiratory Infections at Imperial College London.

**Map disclaimer** The depiction of boundaries on this map does not imply the expression of any opinion whatsoever on the part of BMJ (or any member of its group) concerning the legal status of any country, territory, jurisdiction or area or of its authorities. This map is provided without any warranty of any kind, either express or implied.

**Competing interests** None.

**Patient and public involvement** Patients and/or the public were not involved in the design, or conduct, or reporting, or dissemination plans of this research.

**Patient consent for publication** Not applicable.

**Ethics approval** This study was approved by all Ethical Committee for all sites that participated with this study. For Indonesian sites, this study was approved by Ministry of Health of Indonesia with ethical clearance letter number LB 02.02/2/KE.418/2020. Participants gave informed consent to participate in the study before taking part.

**Provenance and peer review** Not commissioned; externally peer reviewed.

**Data availability statement** Data are available on reasonable request.

**ORCID iDs**
Eva Miranda Marwali http://orcid.org/0000-0002-0135-536X
Malte Kohns Vasconcelos http://orcid.org/0000-0002-6207-9442
Barbara Wanjiru Citarella http://orcid.org/0000-0001-8968-0708

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
