## [Reviewer comments · BMJ Paediatrics Open]

ARTICLE DETAILS

TITLE (PROVISIONAL)	PEDIATRIC COVID-19 MORTALITY: A DATABASE ANALYSIS OF THE IMPACT OF HEALTH RESOURCE DISPARITY
AUTHORS	Marwali, Eva Miranda Kekalih, Aria Yulianto, Saptadi Wati, Dyah K. Rayhan, Muhammad Valerie, Ivy C. Cho, Hwa J Jassat, Waasila Blumberg, Lucille Masha, Maureen Semple, Calum Swann, Olivia Kohns Vasconcelos, Malte Popielska, Jolanta Murthy, Srinivas Fowler, Robert A Guerguerian, Anne-Marie Streinu-Cercel, Anca Pathmanathan, Mohan D. Rojek, Amanda Kartsonaki, Christiana Gonçalves, Bronner P. Citarella, Barbara Wanjiru Merson, Laura Olliaro, Piero L. Dalton, Heidi

VERSION 1 – REVIEW

REVIEWER	Reviewer name: Dr. Conrad Kabali Institution and Country: 2264 Spence Lane, Canada Competing interests: None
REVIEW RETURNED	01-Sep-2022
GENERAL COMMENTS	Page 5, line 15: The correct statistical terms are "univariate" and "multivariable". The terms "bivariate" and "multivariate" are used two outcomes simultaneously or multiple outcomes simultaneously. Page 7, line 42: Even though the samples from countries other than SA and UK are small, it would still be informative to compare mortality between SA and the rest of LMICs, and between UK and the rest of HIC, just to get a sense of how well the two countries represent their regions. Page 7, line 56: Reverse this. It should read 4% vs 1.7%.

REVIEWER	Reviewer name: Dr. Harish PEMDE Institution and Country: Kalawati Saran Children's Hospital, India Competing interests: None
REVIEW RETURNED	25-Apr-2022

GENERAL COMMENTS	Please add a legend for figure 2. Low-middle income group can not account for a risk factor since the comorbidities between country income groups are significantly different. No data on the nutritional status of children in both groups may contribute to the difference in mortality. With a very high number of non-confirmed cases, this study has a weak impact on a policy decision.
--

VERSION 1 – AUTHOR RESPONSE

October 3, 2022

BMJ Paediatrics Open

Dear Editor,

We would like to resubmit the manuscript titled “Paediatric COVID-19 Mortality: A Database Analysis of The Impact of Health Resource Disparity” to be considered for publication in the BMJ Paediatrics Open. We have done some revisions to our paper, based on reviewer’s comments. Below, we also give point by point responses to all the comments from the Editors and our reviewers. As Corresponding Author, I confirm that the manuscript has been read and approved for submission by all the named authors. It is an honor to have a publication in the prestigious BMJ Pediatrics Open.

Sincerely,

Eva Miranda Marwali, MD, PhD

RESPONSES TO BMJ REVIEWER’S COMMENTS

Editor in Chief Comments to Author

1. Title amend to "Paediatric COVID-19 Mortality: A Database analysis of the Impact of Health Resouce Disparity"

Response: Title has been changed accordingly (Page 1)

2. Add Key Messages sections. Please include the key messages of your article after your abstract using the following headings. This section should be no more than 3-5 sentences and should be distinct from the abstract; be succinct, specific and accurate.

What is already known on this topic – summarise the state of scientific knowledge on this subject before you did your study and why this study needed to be done

What this study adds – summarise what we now know as a result of this study that we did not know before

How this study might affect research, practice or policy – summarise the implications of this study

This will be published as a summary box after the abstract in the final published article.

Response: The Key Messages has been added to the manuscript (page 4 after abstract)

What is already known?

Recent systematic reviews identified a potentially higher pediatric mortality per population from COVID-19 in LMIC compared to HIC, but concluded that heterogeneity of published studies limits firm conclusions. Correspondingly, lethality of other acute respiratory infections has been shown to be higher in LMIC.

What this study adds?

By using harmonized data collection tools of a study population of over 12,000 children, this study can directly compare inpatient management and outcomes in HIC and LMIC. Analysis finds higher mortality in LMIC although a lower proportion receive ICU admission and ventilation prior to death. Disparity in access to care and lack of available advanced medical therapies are highlighted and provide areas for collaborative efforts between clinicians, administrators and likely government groups to improve outcomes in LMIC.

How this study might affect policy?

While intensified by the pandemic, a lack of adequate resources to care for children with acute respiratory infections in LMIC is likely a general concern that requires allocation of resources. Reducing the gap in our ability to care for sick children in LMICs versus HICs will inevitably improve global outcomes during both pandemic and inter-pandemic periods.

3. Add a Table listing the countries - number of sites and patients for each country. This should be Table 1

Response: Table 1 has been added (page 16, Table 1).

4. Discussion mentions Supplementary Table 3. This is not in the PDF. Also, it implies that there are also supplementary tables 1 & 2.

Response: Mentioning of supplementary tables are omitted (page 10, line 1)

5. Be cautious in your conclusions

Response: We have changed our sentence in the conclusion of our abstract (page 3):

“Conclusion: Mortality and morbidities were higher in LMICs than HICs, and it may be attributable to differences in patient demographics, complications, and access to supportive and treatment modalities.”

Associate Editor

1. Comments to the Author:

Thank you for your dedicated work on this important and timely article.

We are inclined to accept the work for publication, subject to the minor revisions as recommended by the peer reviewers.

Please consider these recommendations and adjust the manuscript accordingly.

Response: Thank you for the consideration to accept our paper. We have revised our paper according to reviewer's recommendation.

Reviewer: 1

Dr. Conrad Kabali

1. Page 5, line 15: The correct statistical terms are "univariate" and "multivariable". The terms "bivariate" and "multivariate" are used two outcomes simultaneously or multiple outcomes simultaneously.

Response: Changes have been made accordingly (page 3 line 7, page 6 line 7, and page 8 line 9)

2. Page 7, line 42: Even though the samples from countries other than SA and UK are small, it would still be informative to compare mortality between SA and the rest of LMICs, and between UK and the rest of HIC, just to get a sense of how well the two countries represent their regions.

Response: Information regarding comparison of mortality between SA and the rest of LMICs, and between the UK and the rest of HICs, has been added in the narrative of Result and Discussion.

Page 6, the last 2 lines

"Reported mortality in the United Kingdom was 52 children (1.2%), which does not differ significantly ($p = 0.98$) from the rest of HICs. Mortality in South Africa, numbering 268 children (3.5%), differed significantly ($p < 0.001$) from 93 (6.5%) mortality in the rest of LMICs."

Page 8 the last 5 lines

Statistical analysis showed no significant difference of mortality between UK and the rest of HICs. Although mortality in SA was significantly lower than the rest of LMICs, both mortalities from SA and non-SA LMICs were significantly higher than HICs group. Low number of subjects from non-SA LMICs was disproportionate to that of SA, thus conclusion can not be drawn from observed difference in mortality between them.

Page 7, line 56: Reverse this. It should read 4% vs 1.7%.

Response: Changes have been made accordingly

Reviewer: 2

Dr. Antonius Pudjiadi, Universitas Indonesia

Please add a legend for figure 2.

Response: Legend of figure 2 has been enlarged and we also put the legend of the figure in the manuscript document (page 15).

Low-middle income group can not account for a risk factor since the comorbidities between country income groups are significantly different.

No data on the nutritional status of children in both groups may contribute to the difference in mortality.

With a very high number of non-confirmed cases, this study has a weak impact on a policy decision.

Response:

In this registry data, availability for comorbidities is limited, thus we cannot draw much conclusion regarding them. We believed that Low-middle income group can still be accounted as risk factor of mortality.

As for nutritional status, data is also limited and we consider this a limitation of the study.

In this report, nearly 73% of children are confirmed for COVID-19 based on virological examination, but it was true that the rest are non-confirmed cases. We agree that this study may be of limited impact on policy decision due to a degree of missing data . All reviewer's insights have been added in the paragraph about limitation in the end of Discussion (page 10, line 17-20)

“Limitation of this study includes a predominance of patients from one LMICs and one HICs -South Africa and UK, potentially limiting the generalizability of our findings to all countries. In addition, we did not adjust for pandemic era. Inevitably we have missing data for a number of variables, including comorbidities, which limits the effective sample size of analyses examining relationships with patient characteristics and outcomes. Lack of data on nutritional status of children on each group, which may explain disparity between country income groups, was another limitation of the study. Moreover, considerable proportion of non-confirmed cases also limits the impact of this study on public health policy.”